# Molecular Characterization of *ZosmaNRT2*, the Putative Sodium Dependent High-Affinity Nitrate Transporter of *Zostera marina* L.

**DOI:** 10.3390/ijms20153650

**Published:** 2019-07-26

**Authors:** Lourdes Rubio, Jordi Díaz-García, Vítor Amorim-Silva, Alberto P. Macho, Miguel A. Botella, José A. Fernández

**Affiliations:** 1Department of Botánica y Fisiología Vegetal, Campus de Teatinos, University of Málaga, 29071 Málaga, Spain; 2Department Biología Molecular y Bioquímica, Instituto de Hortofruticultura Subtropical y Mediterránea ‘‘La Mayora’’ (IHSM-UMA-CSIC), University of Málaga, Campus Teatinos, 29071 Málaga, Spain; 3Shanghai Center for Plant Stress Biology, CAS Center for Excellence in Molecular Plant Sciences, Shanghai Institutes of Biological Sciences, Chinese Academy of Sciences, Shanghai 200032, China

**Keywords:** *Z. marina* genome, Na^+^-dependent NO_3_^−^ high-affinity transporter, NRT2

## Abstract

One of the most important adaptations of seagrasses during sea colonization was the capacity to grow at the low micromolar nitrate concentrations present in the sea. In contrast to terrestrial plants that use H^+^ symporters for high-affinity NO_3_^−^ uptake, seagrasses such as *Zostera marina* L. use a Na^+^-dependent high-affinity nitrate transporter. Interestingly, in the *Z. marina* genome, only one gene (*Zosma70g00300.1*; *NRT2.1*) is annotated to this function. Analysis of this sequence predicts the presence of 12 transmembrane domains, including the MFS domains of the NNP transporter family and the “nitrate signature” that appears in all members of the NNP family. Phylogenetic analysis shows that this sequence is more related to NRT2.5 than to NRT2.1, sharing a common ancestor with both monocot and dicot plants. Heterologous expression of ZosmaNRT2-GFP together with the high-affinity nitrate transporter accessory protein ZosmaNAR2 (*Zosma63g00220.1*) in *Nicotiana benthamiana* leaves displayed four-fold higher fluorescence intensity than single expression of ZosmaNRT2-GFP suggesting the stabilization of NRT2 by NAR2. ZosmaNRT2-GFP signal was present on the Hechtian-strands in the plasmolyzed cells, pointing that ZosmaNRT2 is localized on the plasma membrane and that would be stabilized by ZosmaNAR2. Taken together, these results suggest that *Zosma70g00300.1* would encode a high-affinity nitrate transporter located at the plasma membrane, equivalent to NRT2.5 transporters. These molecular data, together with our previous electrophysiological results support that ZosmaNRT2 would have evolved to use Na^+^ as a driving ion, which might be an essential adaptation of seagrasses to colonize marine environments.

## 1. Introduction

Seagrasses are the only group of angiosperms that evolved from land plants to complete their life cycle submerged in marine environments. Therefore, the uptake of essential mineral nutrients, such a nitrate or phosphate, occur from a very stable environment that includes high salinity (0.5 M NaCl) and alkaline conditions (pH 8.2). In the case of nitrate, the concentration in seagrass environments is down to 5 µM [1], suggesting that high-affinity transporters developed by terrestrial plants would work in seagrasses, facing both challenges of salinity and alkaline conditions. As vascular plants, these species conserve H^+^-ATPase as the primary pump to energize the plasma membrane [2,3]; nevertheless, in contrast to terrestrial plants that use H^+^ symporters for high-affinity nutrient uptake, seagrasses were the first angiosperms in which high-affinity NO_3_^−^, Pi, and amino acids have been described as Na^+^-dependent mechanisms [4,5,6].

The recent availability of seagrass genomes opens the possibility to identify the genes that encode these singular transport systems [7,8]. The first Na^+^-dependent transport system in plants was characterized in *Zostera marina* L. which mediates high-affinity NO_3_^−^ uptake in mesophyll leaf cells [4], the main organ for nutrient uptake in these plants. This Na^+^-dependent high-affinity NO_3_^−^ transporter also operates in epidermal root cells of this seagrass [5]. *Z. marina*, the most widespread species throughout the temperate northern hemisphere, was also the first seagrass with the genome sequenced [7]. The analysis of its genome shows losses and gains involved in accomplishing morphological and physiological adaptations associated with the reverse evolutionary trajectory of this angiosperm lineage back to the sea [9]. Understanding the molecular mechanisms of these particular Na^+^-dependent high-affinity transport systems to maintain essential nitrate uptake and overcome the high salinity of the marine environment could help to face the problem of land plants growing in increasingly salinized soils.

In vascular terrestrial plants, nitrate transporters were identified and functionally characterized more than 20 years ago (reviewed in [10]). Nitrate uptake is carried out by low-affinity transport systems (LATS) and high-affinity transport systems (HATS), depending on the NO_3_^−^ availability in the soil. The transport activity of these systems demands cellular energy supply and is coupled to the H^+^ electrochemical gradient [11,12]. At least four genes families have been identified to encode plant NO_3_^−^ transporters, the first identified was the *nitrate transporter 1* (*NRT1*) or *peptide transporter* (*PTR*) gene family, recently named the *NPF* family [13]. Members of this family function as the main components of the LATS and operate at high NO_3_^−^ concentrations, except NRT1.1 in *Arabidopsis* that operates as a dual-affinity transporter involved in both HATS and LATS [14]. The second NO_3_^−^ transporter family identified was the *NRT2*, whose members require a nitrate assimilation related protein (NAR2) to transport NO_3_^−^ [15], which is essential for plasma membrane high-affinity NO_3_^−^ transport through NRT2, which functions as HATS under N-limited conditions [16]. The other two NO_3_^−^ transport gene families identified are the chloride channel (CLC) proteins, that are presented in all kingdoms and are related to vacuole NO_3_^−^ accumulation [17], and the *SLAC/SALH* family, that encodes S-type anion channels in guard cells and is involved in stomata closure [18] and contributes to root-to-shoot NO_3_^−^ transport at the root stele [19]. Among all these families, it seems that high-affinity NO_3_^−^ uptake under N-limited conditions belong to NRT2 transporters whose members are targeted to the plasma membrane, except NRT2.7 which has been found in tonoplast in mature seeds [16,20].

In this context, NRT2.1 and its partner protein NAR2.1 are considered as major components of high-affinity NO_3_^−^ uptake under N-limited conditions [10,16,21,22] that operate as a two-component high-affinity nitrate uptake system [23]. In addition, NRT2.2 and NRT2.4 have been involved in HATS. Under N starvation, the expression of *NRT2.4* is rapidly induced and kept at higher levels than *NRT2.1* and *NRT2.2*, which only show a transient increase [24]. All these findings suggest that the interplay between NRT2.1, NRT2.2, and NRT2.4 must be relevant for adaptation to N-limited conditions [25]. Furthermore, *NRT2.5* expression also increases in response to N starvation, similarly to *NRT2.4* but with different spatial-temporal expression patterns [24,26], contributing to NO_3_^−^ uptake by HATS under long-term N starvation [24].

Phylogenetic analysis of *NRT2* family in fully sequenced genomes of vascular plants suggests that most *NRT2* genes developed after the divergence of monocots and dicots, with the only exception of *NRT2.5* orthologues, whose members assemble in the same clade [27]. Less than 20% of gene families are specific in the genome of *Z. marina*, suggesting that colonization of the marine environment should be mostly due to molecular changes of the same gene families rather that the speciation of pre-existing genes [7]. This could be the case of the Na^+^-dependent high-affinity NO_3_^−^ transporter functionally characterized in *Z. marina* mesophyll leaf cells [5].

The aim of this work was to analyze the molecular characteristics of this transport system using an in silico search, protein structure prediction, and phylogenetic analysis of *Z. marina* NRT2 protein sequences. In addition, *NRT2* and *NAR2* cDNAs have been obtained from *Z. marina* leaves and transiently expressed in *Nicotiana benthamiana* to investigate their protein localization.

## 2. Results

### 2.1. Zostera marina High-Affinity NO_3_^−^ Transporter Secuence Identification

To identify high-affinity NO_3_^−^ transporter homologs in the *Z. marina* genome, the NRT2.1 protein sequence from *Arabidopsis thaliana* (accession: O82811) was selected as query in a BLASTp search using Phytozome tools. As result, only one sequence, Zosma70g00300.1, was obtained with a significant identity (55%). This sequence is functionally quoted as high-affinity nitrate transporter 2.1 (ZosmaNRT2) in NCBI and Phytozome databases. The deduced amino acid sequence for ZosmaNRT2 has 517 amino acids residues, including Major Facilitator Superfamily domains (MSF I and II) identified using ScanProsite database, from amino acids 108 to 117 and 321 to 329, respectively (Figure 1). In addition, Nitrate Nitrite Porter motifs (NNP I and II) were also found from amino acids 181 to 187 and 402 to 412, respectively. These motifs have been related to mediate the transport of NO_3_^−^ and NO_2_^−^ in both, prokaryotes and eukaryotes cells. In silico WOLF PSORT server predictions suggest that the most plausible cellular location of ZosmaNRT2 protein would be the plasma membrane. The hydropathy profile and predictions of membrane topology reveal 12 transmembrane (TM) α-helix domains with the amino and carboxylic ends located at the cytosolic face (Figure 1), similar to plant NRT2 proteins. Further, ZosmaNRT2 protein shows a long cytoplasmic loop between TM6 and TM7 that contains more than 30 residues, similar to the NRT2 loop found in different plant species [28].

Comparison of the annotated ZosmaNRT2 protein sequence to other homologous proteins from different phylogenetic groups shows that the most conserved regions of the NRT2 transporters correspond to the predicted transmembrane domains in ZosmaNRT2 (Appendix A). The nitrate signature (NS) motif is found on ZosmaNRT2 TM5, corresponding to the A-G-W/L-G-N-M-G sequence; in addition, the TM11 domain presents a second consensus sequence also considered as an NS (Appendix A). Both NS motifs are two stretches of conserved amino acids that appear in all members of the NNP family [28,29]. Between the TM2 and TM3 domains, a consensus sequence G-x-x-x-D-x-x-G-x-R highly conserved in members of the MFS has been identified [28,30] and in the TM8 domain, another consensus sequence of MFS members is also found [29]. In addition, ZosmaNRT2 contains a conserved motif (F-G-M-R-G-R-L-W) present in NRT2 proteins form photosynthetic eukaryotic organisms [31]. Furthermore, ZosmaNRT2 protein contains two putative phosphorylation sites (S/T-x-R/K) at the C-terminal region (Figure 1 and Appendix A) that are conserved in plants NRT2 proteins NRT2 of plants and play an important role in the regulation of transport activity [28]. All together, these in silico predictions suggest that the ZosmaNRT2 protein is a plasma membrane high-affinity NO_3_^−^ transporter in *Z. marina*.

Next, we used fifty-four sequences for phylogenetic analysis using SeaView4 software (17 sequences from dicots, 30 from monocots, two from *Selaginella moellendorffii*, two from yeasts and three from *Chlamydomonas reinhardtii*). The reconstruction of the phylogenetic tree shows the presence of two different angiosperms clusters, in which monocots and dicots plants are separated (Figure 2). This result indicates that NRT2 proteins from grasses are entirely separated on the phylogenetic lineage from *A. thaliana* NRT2 proteins, except for AtNRT2.5, which is the only one that shares a common ancestor with monocots (Figure 2). The same result is found for ZosmaNRT2 sequence, which has a higher homology with NRT2.5 proteins from dicots than from the other NRT2 proteins included in the phylogenetic analysis, suggesting a common evolutionary origin.

Interestingly, the putative ZosmaNRT2 transporter is phylogenetically more related to the NRT2.5 orthologues from monocots and dicots than to the NRT2.1 proteins. The 517 amino acids sequence of ZosmaNRT2 has an identity between 67%–69% with monocot and dicot NRT2.5 proteins, a similar value (69%) to the identity found with the only NRT2 transporter of the aquatic angiosperm *Egeria densa*. However, the identity with the rest of NRT2 transporters of all angiosperms is lower than 59%, which indicates that ZosmaNRT2 could have an evolutionary relationship close to angiosperms NRT2.5, which would be less coupled to the rest of NRT2 transporters.

### 2.2. The Coding Sequences of ZosmaNRT2 and ZosmaNAR2 were Isolated from Z. marina Leaves mRNA

In order to clone the putative high-affinity transporter *ZosmaNRT2* and the transporter accessory component *ZosmaNAR2* (*Zosma63g00220.1*) cDNAs, we designed two gene specific primers (Appendix A). Using RNA extracted from the *Z. marina* leaves as template, two fragments of 1554 and 633 bp were obtained by RT-PCR, which corresponded to *ZosmaNRT2* and *ZosmaNAR2*, respectively.

The sequencing results showed an open reading frame for 517 amino acids in ZosmaNRT2.1. Analysis of the gene structure showed two exons and one intron, unlike grass NRT2 genes that do not contain introns [27]. This would indicate an ancient divergence of *ZosmaNRT2* regarding monocot *NRT2.5*. On the other hand, *ZosmaNAR2* contains two exons and one intron like grasses and *A. thaliana*. The encoded protein of ZosmaNAR2 contains 210 amino acids and shares a 45% amino acid sequence similarity with AtNAR2.1. Unlike the single transmembrane domain found in AtNAR2.1, ZosmaNAR2 is predicted to contain two transmembrane domains, TM1 from amino acid 5 to 27 and TM2 from amino acid 181 to 203, respectively.

### 2.3. ZosmaNAR2 Stabilizes the Location of ZosmaNRT2 at the Plasma Membrane

As indicated above, ZosmaNRT2 contains 12 transmembrane domains and is predicted as a plasma membrane protein based on in silico analysis. To investigate in vivo subcellular localization, a green fluorescent protein (GFP) was fused in frame to the C-terminal of ZosmaNRT2 driven by the constitutive cauliflower mosaic virus 35S promoter (*35S::ZosmaNRT2-GFP*). We also generated a construct with the transporter accessory protein ZosmaNAR2 under 35S promoter (*35S::ZosmaNAR2*). These constructs were transiently expressed in *N. benthamiana* leaves and further analyzed. Expression of *35S::ZosmaNRT2-GFP* showed a weak signal, although above that of the empty vector (Figure 3A–D). Interestingly, coexpressing the high-affinity nitrate transporter *35S::ZosmaNRT2-GFP* together with the accessory protein (*35S::ZosmaNAR2*) cause a drastic increase in GFP signal (Figure 3E,F), indicating that ZosmaNAR2 is important for the accumulation of ZosmaNRT2.

In order to quantify GFP fluorescence, RAW integrated density of green pixels per total area was calculated as described in methods. Coexpression of *35S::ZosmaNRT2-GFP* and *35S::ZosmaNAR2* evokes four-fold higher relative GFP fluorescence than the observed in epidermal *N. benthamiana* expressing *35S::ZosmaNRT2-GFP* without *35S::ZosmaNAR2* (Figure 4). Interestingly, GFP signal was mainly detected at the cell periphery pointing that ZosmaNRT2-GFP was localized at the plasma membrane (Figure 3).

To verify the plasma membrane localization of ZosmaNRT2-GFP, *N. benthamiana* leaf cells coexpressing *35S::ZosmaNRT2-GFP* and *35S::ZosmaNAR2* were plasmolyzed and observed under the confocal microscope. After incubation in 0.8 mM Mannitol, plasmolyzed cells shows the Hechtian-strands labelled by GFP fluorescence (Figure 5) according to a plasma membrane localization of ZosmaNRT2-GFP. Hechtian-strands labelled by GFP fluorescence were not observed in single *35S::ZosmaNRT2-GFP* heterologous expression assays. These results support the in silico prediction for ZosmaNRT2 localization and its stabilization at the plasma membrane by NAR2, which fits the function of ZosmaNRT2 as a NO_3_^−^ transporter.

## 3. Discussion

High-affinity NO_3_^−^ transporters belonging to the NRT2 family show a ubiquitous distribution and are widely represented in the genomes of plants, fungi, microalgae, and bacteria [22]. In the case of the seagrass *Z. marina* genome only one sequence, *Zosma70g00300.1* has been identified and is annotated as a NO_3_^−^ high-affinity transporter related to NRT2. Many studies have shown that these NRT2 transporters are included in the MFS superfamily (major facilitator superfamily), one of the main groups of transporters that are characterized by 12 transmembrane domains in the protein sequence [28]. Our detailed in silico analysis of ZosmaNRT2 predicts a protein topology with the same number of transmembrane domains (Figure 1). In addition, the consensus sequences, highly conserved among members of the MFS superfamily, are also present in the ZosmaNRT2 protein. On the other hand, the ZosmaNRT2 protein also contains the two conserved “NS sequences” motifs, a unique trait of the NNP (nitrate nitrite porter) family of eukaryotes and prokaryotes, since NS sequences are not found in other members of the MFS superfamily [28]. Furthermore, the two putative phosphorylation sites (S/T-x-R/K) that are conserved in plant NRT2 proteins have also been found in the sequence of ZosmaNRT2 (Appendix A), motifs that play an important role in the regulation of the transport activity of the NNP [28]. It is known that members of the NNP family mediate both NO_3_^−^ and NO_2_^−^ transport [28]; within this group of NNP transporters are included the high affinity NO_3_^−^ transporters of *Aspergillus nidulans* (NRTA), NRT2 proteins of higher plants, NARK and YNT1 transporters of prokaryotes and *Hansenula polymorpha* yeast, respectively [32]. In all cases, they are very similar functional carriers, with affinity values for nitrate in the μM range. All these characteristics shared with the MFS superfamily and members of the NNP family suggest that ZosmaNRT2 protein could also be a high-affinity NO_3_^−^ transporter in *Z. marina*.

Based on the hydropathy profile, one of the most conserved characteristics in the topology of NNP nitrate transporters is the extensive central loop at the cytosolic side that connects the TM6 and TM7 transmembrane domains [28]. In the case of prokaryotes, the length of the loop is lower than in fungal nitrate transporters, which can reach up to 90 amino acids, and up to 40 amino acids in algae and higher plants [28]. This structure is also present in the predicted topology for ZosmaNRT2 with a central cytosolic loop of 31 amino acids (Figure 1). Furthermore, in all members of the NNP family, it has been experimentally shown that N and C-terminal domains and the central loop fall on the cytosolic side of the plasma membrane [33,34,35]. This feature is also preserved in the proposed topology for ZosmaNRT2 (Figure 1). In ZosmaNRT2, the C-terminal tail reaches an extension of 69 amino acids and lacks secondary structure like the unstructured tails of monocot and dicot NNP transporters, which show particularly long C-terminal domains (~70 amino acids) and drastically differ from the C-terminal domains almost absent from fungal nitrate transporters NRTA and YNT1 [28]. This discrepancy between the C-terminal tails of the fungal nitrate transporters, which do not require the auxiliary protein NAR2 to transport nitrate, indicates that this C-terminal domain could play an important role in the interaction with NAR2 and the regulation of transport activity [29]. Accordingly, it has been found that Ser463, located at the C-terminal domain of the HvNRT2 protein of barley, is required for the interaction with NAR2 [36]. This residue, also found in ZosmaNRT2 (Figure 1), is conserved in algae and monocots but not in dicots. Then, it is likely that ZosmaNRT2 also requires the interaction of NAR2 to mediate NO_3_^−^ transport. Our data of *N. benthamiana* coexpression assays suggest that the presence of NAR2 is important to stabilize ZosmaNRT2 protein at the plasma membrane (Figure 3). In addition, the amount of ZosmaNRT2 protein in the plasma membrane, revealed as GFP signal, is four-fold higher when coexpressed together with the ZosmaNAR2 protein in the same cell (Figure 4). This observation has been previously reported for AtNRT2.1, wherein plasma membrane localization increased when AtNAR2.1 was present, suggesting a two-component (AtNRT2.1-AtNAR2.1) NO_3_^−^ high-affinity uptake system [23]. Interestingly, this interaction has been also reported for AtNRT2.5, indicating that NAR2 is essential for AtNRT2.5 targeting to the plasma membrane where both proteins formed an oligomeric complex of 150 KDa [37].

The reconstruction of the phylogenetic tree by homology between amino acid sequences of the NRT2 proteins of plants, fungi, and microalgae shows that these proteins are grouped in several clusters: a group of NRT2 paralogs in dicots and another in monocots (Figure 2; green and blue clades, respectively); a clade that mainly groups NRT2.5 orthologues from dicot and monocot species (orange); two possible *S. moellendorffii* paralogs (pink) that appears after the divergence between NRT2.5 and the other NRT2 transporters, except NRT2.7 (Figure 2). This phylogenetic tree indicates that the only high-affinity nitrate transporter found in the *Z. marina* genome, ZosmaNRT2, shares a common ancestor with the NRT2.5 proteins of monocots and dicots, but is completely separated from the phylogenetic lineage of the rest of the NRT2 proteins of *Arabidopsis* and dicots (Figure 2). Additionally, the protein sequence of ZosmaNRT2 shares an identity of about 70% homology with NRT2.5 of monocot and dicot species, whereas it shares less than 59% homology with the rest of the NRT2 transporters. Consequently, this relationship suggests a common evolutionary origin of ZosmaNRT2 and the NRT2.5 transporters from monocots and dicots, which could imply similar function. The same result has been observed in an exhaustive phylogenetic analysis of the *NRT2* genes, in which it was concluded that the *NRT2.1*, *2.2*, *2.3*, *2.4*, and *2.6* genes from monocots are completely separated from the dicot species, except for *NRT2.5*, which is the only *NRT2* gene that has clear orthologues among dicot and monocot species, and therefore the same evolutionary origin [27].

Plett et al. [27] proposes that the high-affinity NO_3_^−^ transporter NRT2 existed in higher plants even before the divergence of monocots and dicots. Prior to the separation of these species, there was an event of gene duplication of the NRT2 family, giving rise to pairs of NRT2 co-orthologous genes. Then, a strong functional radiation was generated, especially in dicots, which led to the appearance of multiple genetic copies with different functions (*NRT2.1*, *2.2*, *2.3*, *2.4*, *2.6*), including different physiological functions for the same paralogs as occurs between *AtNRT2.1* and *AtNRT2.2* [38]. Interestingly, the *ZosmaNRT*2 gene has an intron unlike the NRT2 genes of grasses that lack introns. This would indicate a genomic divergence after the appearance of the seagrasses, likely coupled to the functional divergence of this transport system in marine angiosperms, because has experienced an important functional specialization to use Na^+^ as a driving ion instead of H^+^.

The functional radiation and appearance of new versions of NRT2 transporters seems not be the case of *Z. marine* that only has one NRT2 transporter in their genome. This marine plant is characterized by using Na^+^ as a driving ion to incorporate NO_3_^−^ and by growing in oligotrophic media, whose availability of nitrate in the water column is around 5 μM and the oscillations in NO_3_^−^ levels are very moderate [39]. However, the concentration of nitrate in the soil can change rapidly due to the action of various biotic and abiotic factors, causing enormous spatial and temporal heterogeneity in the availability of NO_3_^−^ for terrestrial plants [40]. Therefore, it is hypothesized that terrestrial plants have had to supply several versions of NRT2 and NRT1 transporters with different affinities for NO_3_^−^ and different degrees of regulation to quickly respond to soil NO_3_^−^ fluctuations [37]. On the contrary, the unique NRT2 transport system found in *Z. marina* might be able to efficiently cover the acquisition of NO_3_^−^ in a medium where its availability is both, very low and very stable, showing an affinity for NO_3_^−^ (*K_m_* 2.31 ± 0.78 μM, [4]) substantially higher than those described in many other terrestrial angiosperms (6–20 μM, [40]), that ensures the uptake of NO_3_^−^ in marine environment. This function of ZosmaNRT2 is similar to NRT2.5 transporters in *Arabidopsis*, which are involved in NO_3_^−^ uptake under prolonged nitrogen starvation or when the NO_3_^−^ concentrations in the medium are very low [24,41]. In this sense, Lezhneva et al. [24] observed that, after a long period of NO_3_^−^ deficiency, the expression of AtNRT2.5 is strongly induced and becomes the predominant high-affinity transporter in *Arabidopsis* roots, while the expression of the rest of the *AtNRT2* transporters remains at low levels. Then, the expression of *AtNRT2.5* decreases after providing NO_3_^−^ to the medium. Furthermore, its low value of *K_m_* makes it an adequate transporter to guarantee NO_3_^−^ acquisition at the trace amounts in root environments [24]. A similar behaviour was observed in corn where it was found that the transporter ZmNRT2.5 was the only high-affinity NO_3_^−^ transporter that was significantly expressed when the plants were grown at a low NO_3_^−^ concentration [42]. In addition, the presence of NRT2.5 is not exclusive to root cells. In the leaves of adult *Arabidopsis* plants, the amount of *AtNRT2.5* mRNA is increased by 20% and has a maximum expression in senescent leaves and during N deficiency [24,26]. Thus, NRT2.5 transporters could be also expressed in leaves as ZosmaNRT2 in *Z. marina* although, presumably, the function of NRT2.5 in the leaves of dicots is remobilization of NO_3_^−^ from intracellular nitrate deposits to the rest of the plant via phloem [24].

The high equivalence of ZosmaNRT2 to NRT2.5 proteins reported here indicates that *ZosmaNRT2* has been being incorrectly annotated as *ZosmaNRT2.1* in genomic databases. *NRT2* genes’ wrong annotation has been previously suggested [27]. Taken together, our detailed phylogenetic analysis indicates the presence of only one *NRT2* gene in the *Z. marina* genome, which is most related to *AtNRT2.5*. This finding supports the idea that *ZosmaNRT2* encodes the high-affinity NO_3_^−^ transporter operating at the *Z. marina* cell plasma membrane, which has evolved to use Na^+^ as the driving ion. The isolation of both genes for the first time in a seagrass, opens the possibility of further analysis, which would provide important insights for NO_3_^−^ uptake in plants under alkaline, NO_3_^−^-limited, and salinized environments.

## 4. Materials and Methods

### 4.1. Plant Materials and Growth Conditions

*Zostera marina* L. plants were collected from Cádiz Bay (36°29′25.9″ N, 6°15′49.0″ W, Spain) and transported within 2 h to the laboratory. Once in the lab, surface epiphytes and older leaves were removed from the plants. Then, plants were maintained in plexiglass containers filled with aerated natural seawater (NSW) at 15 °C and under a photon flux ratio of 150 μmol m^−2^ s^−1^ (16/8 h light/dark photoperiod). NSW was renewed every three days. Only healthy leaves of *Z. marina* plants were used for molecular assays. *Nicotiana benthamiana* D. seeds were germinated and grown in soil containing a mixture of organic substrate and vermiculite (4:1 *v*/*v*). Seedlings were placed in grown chamber under control conditions: 23 °C ± 1 °C, with a photoperiod 16/8 h light/dark (120 µmol photon m^−2^ s^−1^) and irrigated with tap water twice per week. Leaves of four-week-old plants were used for molecular assays.

### 4.2. Sequence and Phylogenetic Analysis

BLASTp search using Phytozome v12.1 search tool [43] was used to identify putative high-affinity nitrate transporter proteins using *A. thaliana* NRT2.1 (AT1G08090) as query. Multiple sequence alignments (MSA) on the entire sequences including plants, algae, fungi, and bacteria NRT2 proteins were performed by MultAlin alignment tool (http://multalin.toulouse.inra.fr) [44]. Neighbor-joining method [45] was used to generate phylogenetic tree using the bootstrap re-sampling analysis (1000 bootstrap replicates) included in the SeaView4 program [46]. FigTree v1.4.4 software (http://tree.bio.ed.ac.uk/software/figtree/) was used for visualization.

The TMHMM 2.0 server [47] was used for predictions of putative transmembrane domains (TMDs). The online PROTTER application was used for graphic representation of protein topology. The ScanProsite tool [48] was used to search for patterns and conserved protein domains. Presence of signal peptides in the N-terminal region and subcellular localization were obtained using the Protein Prowler Prediction server [49] and the WOLF PSORT server [50], respectively.

### 4.3. Obtaining NRT2 and NAR2 cDNAs from Z. marina Leaves.

Total RNA was extracted from green leaves of *Z. marina* plants according to method described by [51] and treated with *TURBO DNA-free™* (Ambion, Carlsband, CA, USA). RNA was quantified using a NanoDrop™ One^C^ Spectrophotometer (Thermo Fisher Scientific, Wilmington, USA) and the integrity of RNA was qualitatively checked on a 1% agarose gel. One µg of total RNA was used for cDNA synthesis using *iScript^TM^ cDNA Synthesis* kit (Bio-Rad, Hercules, California, CA, USA). NRT2 and NAR2 sequences were amplified by PCR using the reverse transcription products as template, the proofreading TaKaRa La Taq DNA Polymerase (TaKaRa, Tokyo, Japan), and specific primer pairs containing BamHI and XbaI restriction sites (Appendix A). The nucleotide sequence was verified by sequencing.

### 4.4. Molecular Cloning

cDNA sequences were inserted into pGEM^®^-T Easy vector (Promega, Wisconsin, WI, USA) through ligation by hanging thiamines and following the manufacturer’s instruction to transform *E. coli* DH5α^TM^ strain (ThermoFisher, Wilimington, DE, USA) for propagation. To construct entry clones, cDNA sequences were amplified without stop codon using proofreading TaKaRa La Taq DNA Polymerase and primer sequences included Gateway^®^ Technology recombination sites (Appendix A). Once purified, amplified fragments were combined with donor vectors pDONR^TM^/Zeo (Invitrogen, Grand Island, New York, NY, USA), using BP Clonase II (Invitrogen) in BP recombination reaction. The resulting clones were verified by diagnostic PCR, restriction analysis, and sequencing. Subsequently, destination clone for *ZosmaNRT2* cDNA was prepared by Gateway LR reaction using pDONR^TM^/Zeo-NRT2, pGWB5 binary expression vector and LR Clonase II enzyme mix (Invitrogen) producing a *ZosmaNRT2-GFP* construct under the control of the 35S promoter (*35S::ZosmaNRT2-GFP*). In the same way, *ZosmaNAR2* was cloned into pGWB14 binary expression vector, obtaining a ZosmaNAR2-HA construct that carries NAR2 fused in-frame to hemagglutinin epitope tag in the C-terminal under the control of the 35S promoter (*35S::ZosmaNAR2*). Both plasmids were then used to transform *Agrobacterium tumefaciens* strain GV3101::pMP90 by electroporation.

### 4.5. Transient Expression in N. benthamiana Leaves and Subcelular Localization

For transient expression in *N. benthamiana*, four-week-old leaves were infiltrated at the abaxial side by *Agrobacterium* carrying pGWB5-NRT2:GFP and/or pGWB14-NAR2:HA plasmids and p19 strain [52]. *Agrobacterium* cultures were grown overnight in LB medium containing rifampicin (50 µg/mL), gentamycin (25 µg/mL), and kanamycin (50 µg/mL). Cells were then harvested by centrifugation (15 min, 3000× *g* in 50 mL falcon tubes), pellets were diluted in agro-infiltration solution (10 mM MES pH 5.6, 10 mM MgCl_2_ and 1 mM acetosyringone) and incubated for 2 h in dark conditions at room temperature. 

Agrobacterium strains were infiltrated at 0.4 (OD_600_) for the constructs and 0.2 (OD_600_) for the p19 strain. An *Agrobacterium* strain harbouring an empty vector was used as a negative control or to equal the final optical density to approximately 1 in all infiltration experiments. After infiltration, all plants were kept in the culture chamber for 2 days before confocal analysis.

Lower epidermis leaf confocal images were obtained using a Leica TCS SP5 II confocal microscope equipped with a 488 nm argon laser for GFP. Trials for plasma membrane localization consisted of a leaf plasmolysis treatment using 0.8 M mannitol for 20 min before confocal image capture [53].

### 4.6. Image Processing and Fluorescence Intensity Quantification

Leica LAS AF Lite platform and Java-based image-processing FIJI software (National Institutes of Health) were used for images process. Fluorescence intensity was quantified by ImageJ software. RGB images (1024 × 1024 pixels) were converted to an eight-bit images based on a grayscale (0–255) to create a binary mask in the green channel. This mask allows the selection from the original image to measure fluorescence in regions of interest. For this, the bandpass FFT_Filter plugin was applied to filter out large structures down to 40 pixels (shading correction) and small structures up to 10 pixels (smoothing) of the specified size by Gaussian filtering in Fourier space. Then, an auto threshold level was applied to the resultant image using Li method to segment the image; pixel values under threshold level are considered noise and not analyzed. “White object on black background” options was selected to convert into white the pixels with values above the threshold level and “Ignore black” option was used to turn these pixels into cero of the grayscale in the binary image. The defined areas by binary masks were overlaid to the original RGB images for green fluorescence quantification. Measurements consisted of the RAW integrated density of pixels per total selected area in each image. Fluorescence values are presented as mean ± SD of four images from three independent experiments. Data were analyzed by one-way ANOVA and post hoc Tukey test, using SPSS Statistics, version 21. The significance level was set at *p* < 0.05.

## Figures and Tables

**Figure 1 ijms-20-03650-f001:**
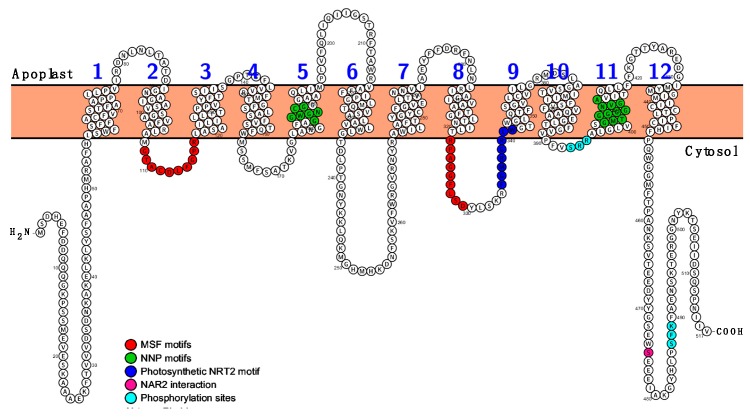
Predicted transmembrane topology for ZosmaNRT2 protein sequence. The 2D model was obtained by using the PROTTER program and TMHMM 2.0 for hydropathy. Conserved amino acids corresponding to each signature motif are colored as indicated: Mayor Facilitator Superfamily motifs (MFS I and II) in red, Nitrate Nitrite Porter motifs (NNP I and II) in green, and blue for the Photosynthetic NRT2 motif. S/T-x-R/K motifs, found in the majority of vascular plant, are colored in light blue, including Ser-393 and Ser-487, the phosphorylation sites. Ser-475, highlighted in dark pink, could be involved in NAR2 interaction (discussed in the text).

**Figure 2 ijms-20-03650-f002:**
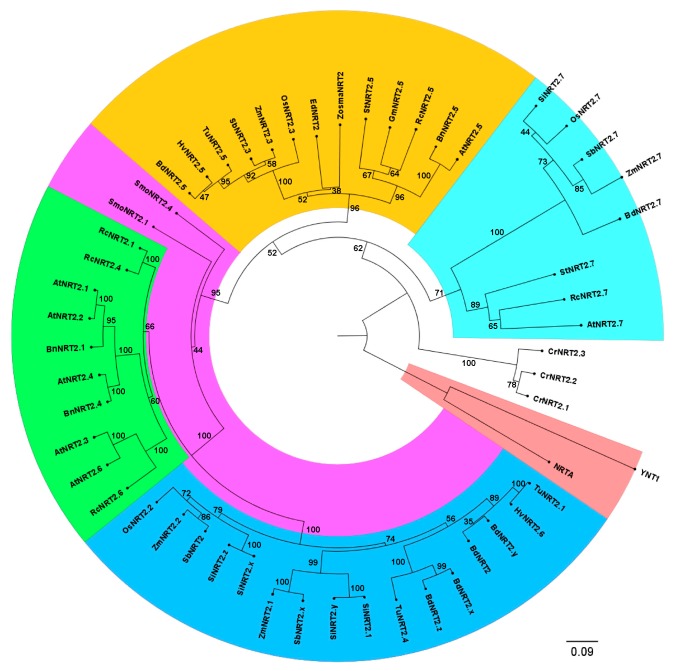
Phylogenetic reconstruction of the NRT2 protein family. Unrooted neighbor-joining tree based on a set of 54 amino acid sequences of the NRT2 proteins identified as the best GenBank and Phytozome BLAST hits using representative members from each taxonomic group. The percentage of trees in which the associated taxa clustered together is shown next to the branches (1000 bootstrap replicates). The accession numbers of each protein in the NCBI included in the analysis were: *Zostera marina* (ZosmaNRT2, KMZ59016); *Arabidopsis thaliana* (AtNRT2.1, O82811; AtNRT2.2, Q9LMZ9; AtNRT2.3, AED97376; AtNRT2.4, Q9FJH8; AtNRT2.5, Q9LPV5; AtNRT2.6, Q9LXH0; AtNRT2.7, Q9LYK2); *Brassica napus* (BnNRT2.1, XP_013729508; BnNRT2.4, XP_013665864; BnNRT2.5, XP_013657329) *Ricinus communis* (RcNRT2.1, XP_002523687; RcNRT2.4, XP_002523688; RcNRT2.5, XP_002527899; RcNRT2.6, XP_002523689; RcNRT2.7, XP_002524664); *Solanum tuberosum* (StNRT2.5, XP_006367151; StNRT2.7, XP_006357155); *Oryza sativa* (OsNRT2.2, XP_015623596; OsNRT2.3, XP_015628524; OsNRT2.7, XP_015636108); *Sorghum bicolor* (SbNRT2, XP_002453159; SbNTR2.x, XP_002453159; SbNRT2.3, XP_002456219; SbNRT2.7, XP_002455791); *Triticum urartu* (TuNRT2.1, EMS65311; TuNRT2.4, EMS46096; TuNRT2.5, EMS50263); *Zea mays* (ZmNRT2.1, XP_008645163; ZmNRT2.2, NP_001105195; ZmNRT2.3, XP_008656795; ZmNRT2.7, AQK44570); *Hordeum vulgare* (HvNRT2.5, ABG20828; HvNRT2.6, ABG20829); *Brachypodium distachyon* (BdNRT2, XP_003572454; BdNRT2.x, XP_003570801; BdNRT2.y, XP_003572550.1; BdNRT2.z, XP_003572590; BdNRT2.5, XP_003569637; BdNRT2.7, XP_003566766.2); *Glycine max* (GmNRT2.5, XP_003531994); *Setaria italica* (SiNRT2.x, XP_004952143; SiNRT2.y, XP_004952138; SiNRT2.z, XP_004952140; SiNRT2.1, XP_022679328; SiNRT2.7, XP_004968955); *Egeria densa* (EdNRT2, BAK51923); *Selaginella moellendorffii* (SmoNRT2.1, XP_002993278; SmoNRT2.4, XP_002966266); *Chlamydomonas reinhardtii* (CrNRT2.1, XP_001696789; CrNRT2.2, Q39609.2; CrNRT2.3, CAD60538.1); *Hansenula polymorpha* (YNT1; CAA93631); *Aspergillus nidulans* (NRTA, XP_658612). The phylogenetic tree was obtained through SeaView4 program and visualized by the FigTree v1.4.4 software. Different colors are used for each clade. The scale bar represents a 0.09 estimated amino acid substitution per residue.

**Figure 3 ijms-20-03650-f003:**
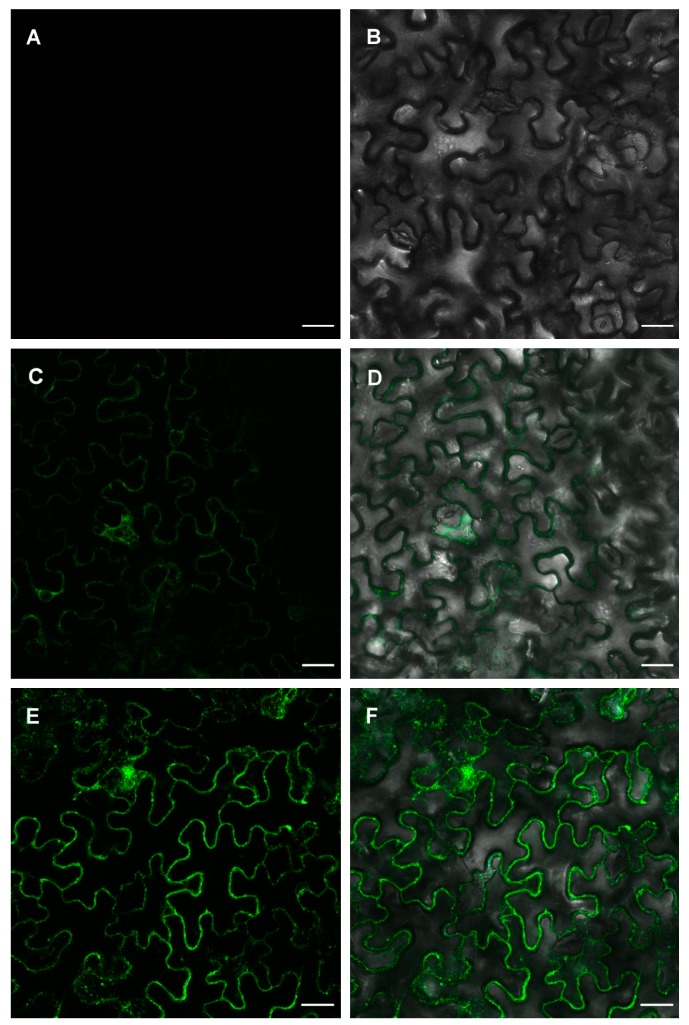
Transient expression of ZosmaNRT2 and ZosmaNAR2 in *N. benthamiana* leaves using *Agrobacterium* infection. Fluorescence (**A**) and merge of fluorescence and bright field (**B**) images of epidermal leaf cells transiently transformed with empty vector pGWB1 used as the control. Fluorescence (**C**) and merge of fluorescence and bright field (**D**) images of epidermal leaf cells transiently transformed with *35S::ZosmaNRT2-GFP*. Fluorescence (**E**) and merge of fluorescence and bright field (**F**) images of epidermal leaf cells transiently cotransformed with *35S::ZosmaNRT2-GFP* and *35S::ZosmaNAR2*. Each confocal image is a representative picture of ten independent assays. Bars: 25 µm.

**Figure 4 ijms-20-03650-f004:**
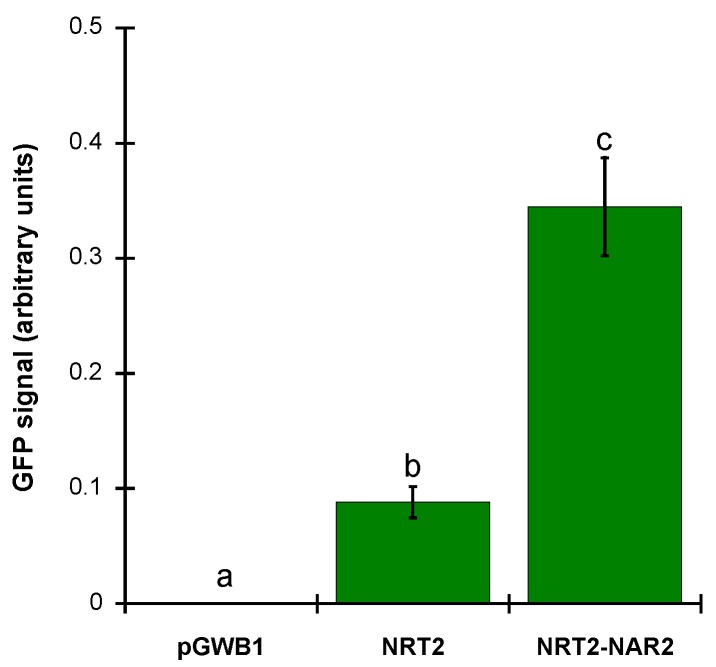
ZosmaNRT2-GFP fluorescence intensity. GFP signal level was calculated as RAW integrated density of fluorescence images of *N. benthamiana* epidermal leaf cells transiently transformed with empty vector pGWB1 (Control), *35S::ZosmaNRT2-GFP* (NRT2) or coexpressing *35S::ZosmaNRT2-GFP* and *35S::ZosmaNAR2* (NRT2-NAR2). ImageJ was used to quantify the GFP signal levels. Data are mean ±SD of four independent experiments. Different letters indicate significant differences (Tukey Test, *p* < 0.0001).

**Figure 5 ijms-20-03650-f005:**
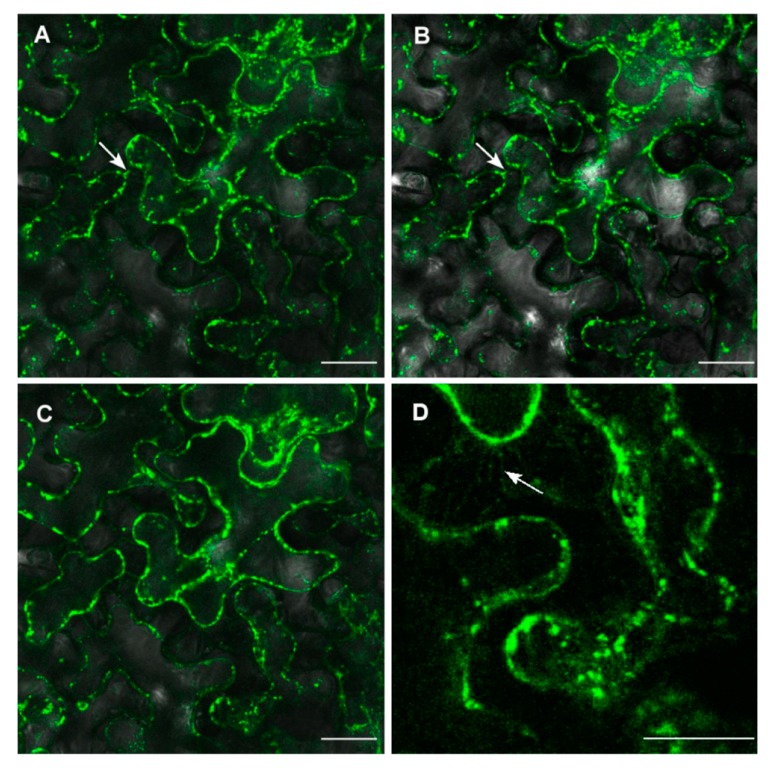
Cellular localization of ZosmaNRT2 transiently expressed in *N. benthamiana* leaves. Confocal images of epidermal leaf cells coexpressing 35S::ZosmaNRT2-GFP and 35S::ZosmaNAR2 were taken after 25 min (**A**), 26 min (**B**), and 28 min (**C**,**D**) of 0.8 M Mannitol treatment. Arrows indicate the GFP signal in the Hechtian-strands of the plasma membrane. Images are representative of ten independent trials. Bars: 25 µm (**A**–**C**) and 10 µm (**D**).

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
