# Peer review of "Molecular Characterization of ZosmaNRT2, the Putative Sodium Dependent High-Affinity Nitrate Transporter of Zostera marina L."

_ijms, 2019, doi:10.3390/ijms20153650_

Reviewer 1 Report

Dear authors,

in this manuscript you are trying to show that ZosmaNRT2 is a high-affinity nitrate transporter from Zostera marina, as you entitled your manuscript.

Unfortunately, you deliver almost no evidence for this hypothesis. First, you identified this sequence from the whole genome sequence of Zostera, and it shows similarities to other plant NRTs. This indicates that your protein may have this function, but needs experimental support, which is almost completely missing. Even the bootstrap support of almost all the branches of your phylogenetic tree are so low (most of them below 50%!), that the tree itself is highly unreliable.

As experimental evidence you expressed ZosmaNRT2-GFP transiently in tobacco together with the accessory protein ZosmaNAR2. In these pictures (Fig. 3) it seems that ZosmaNRT2 localizes to the plasma membrane. Instead of using a membrane marker for microscopy or membrane extraction and analysis to support this observation, you argue with the binding of ZosmaNRT2 to Hechtian strands of plasmolyzed cells. On the one hand side, this argumentation is highly unusual and debatable. On the other hand, the fluorescence of these Hechtian strands you show in these pictures (Fig. 5) is very weak and also controls are missing to confirm your observation.

Additionally, the conclusions you draw from your experiments with the accessory protein ZosmaNAR2 are far too speculative. After cotransformation of this protein with ZosmaNRT2-GFP you observe higher GFP intensities. You argue that this comes from increased accumulation due to ZosmaNAR2 and try to confirm this by pixel counting (Fig. 4). But you have not a single clue for this! The higher GFP intensities may stem from dozens of reasons. You can´t prove that ZosmaNAR2 is even expressed. Therefore, experiments with a ZosmaNAR2-RFP fusion would have been far more informative.

Altogether, except for the putative plasma membrane localization of ZosmaNRT2 you have no experimental indication for your assertion, that this protein is really a transporter.

Author Response

Dear authors,

in this manuscript you are trying to show that ZosmaNRT2 is a high-affinity nitrate transporter from Zostera marina, as you entitled your manuscript.

Unfortunately, you deliver almost no evidence for this hypothesis. First, you identified this sequence from the whole genome sequence of Zostera, and it shows similarities to other plant NRTs. This indicates that your protein may have this function, but needs experimental support, which is almost completely missing. Even the bootstrap support of almost all the branches of your phylogenetic tree are so low (most of them below 50%!), that the tree itself is highly unreliable.

Thank for your comments. After consider all of them, we have decided to change the manuscript tittle by “Molecular characterization of ZosmaNRT2, the putative sodium dependent high-affinity nitrate transporter of Zostera marina L.” As stated, the aim of this work was the molecular characterization of the only sequence annotated as a high-affinity NO3- transporter in the Zostera marina genome. We consider that presented data are original and could be relevant to be publish at the IJMS Special Issue "Uptake and Compartmentalisation of Mineral Nutrients in Plants", since for the first time in a marine plant, both genes, ZosmaNRT2 and its accessory protein ZosmaNAR2 have been isolated and characterised at the molecular level.

Taking into account your first recommendation we have added more sequences (54 in total) to build the phylogenetic tree (New Figure 2), obtaining higher probability in most of the branch (only 5 of the 50 branch are between 35 and 50%). Despite to the addition of 19 new sequences, similar results have been obtained, that is, ZosmaNRT2 protein is clustered into NRT2.5 homologues, which are separated from the rest of NRT2 transporters (New Figure 2).

As experimental evidence you expressed ZosmaNRT2-GFP transiently in tobacco together with the accessory protein ZosmaNAR2. In these pictures (Fig. 3) it seems that ZosmaNRT2 localizes to the plasma membrane. Instead of using a membrane marker for microscopy or membrane extraction and analysis to support this observation, you argue with the binding of ZosmaNRT2 to Hechtian strands of plasmolyzed cells. On the one hand side, this argumentation is highly unusual and debatable. On the other hand, the fluorescence of these Hechtian strands you show in these pictures (Fig. 5) is very weak and also controls are missing to confirm your observation.

ZosmaNRT2 and ZosmaNAR2 transient expression assays have been repeated 10 times and similar results are always obtained. This experimental approximation is found in plant research literature to identify membrane protein localization, specially, in the case of plasma membrane transporters.

1.     Baker et al., 2016. Sucrose Transporter ZmSut1 Expression and Localization Uncover New Insights into Sucrose Phloem Loading. PLANT PHYSIOLOGY 172: 1876–1898. doi.org/10.1104/pp.16.00884

2.     Chanroj et al., 2013. K+ Transporter AtCHX17 with Its Hydrophilic C Tail Localizes to Membranes of the Secretory/Endocytic System: Role in Reproduction and Seed Set. MOLECULAR PLANT 4: 1226–1246. doi.org/10.1093/mp/sst032

3.     Li et al., 2016. Identification of a Stelar-Localized Transport Protein That Facilitates Root-to-Shoot Transfer of Chloride in Arabidopsis. PLANT PHYSIOLOGY 170: 1014–1029. doi.org/10.1104/pp.15.01163 

4.     Li et al., 2017. AtNPF2.5 Modulates Chloride (Cl) Efflux from Roots of Arabidopsis thaliana. FRONTIERS IN PLANT SCIENCE 2013. doi.org/10.3389/fpls.2016.02013

5.     Schmitt et al., 2008. Immunolocalization of solanaceous SUT1 proteins in companion cells and xylem parenchyma: new perspectives for phloem loading and transport. PLANT PHYSIOLOGY 148: 187–199. doi.org/10.1104/pp.108.120410.

6.     Wen et al., 2017. Maize NPF6 proteins are homologs of Arabidopsis CHL1 that Are selective for both nitrate and chloride. PLANT CELL 29: 2581–2596 doi.org/10.1105/tpc.16.00724

On the other hand, ZosmaNRT2-GFP signal was present on the Hechtian-strands only in ZosmaNRT2-ZosmaNAR2 co-expression assays. This result is consistent with the low ZosmaNRT2-GFP signal in the absence of NAR2 (Figure 4) and points to a role of NAR2 for the NRT2 stabilization at the plasma membrane. The following comment has been included in the main text (page 7, lines 224-227):

“Hechtian-strands labelled by GFP fluorescence were not observed in single 35S::ZosmaNRT2-GFP heterologous expression assays (Data not shown). These results support the in silico prediction for ZosmaNRT2 localization and its stabilization at the plasma membrane by NAR2, which would fit to the function of ZosmaNRT2 as a NO3- transporter”

Additionally, the conclusions you draw from your experiments with the accessory protein ZosmaNAR2 are far too speculative. After cotransformation of this protein with ZosmaNRT2-GFP you observe higher GFP intensities. You argue that this comes from increased accumulation due to ZosmaNAR2 and try to confirm this by pixel counting (Fig. 4). But you have not a single clue for this! The higher GFP intensities may stem from dozens of reasons. You can´t prove that ZosmaNAR2 is even expressed. Therefore, experiments with a ZosmaNAR2-RFP fusion would have been far more informative.

Altogether, except for the putative plasma membrane localization of ZosmaNRT2 you have no experimental indication for your assertion, that this protein is really a transporter.

Accordingly to the scope of the Special Issue "Uptake and Compartmentalisation of Mineral Nutrients in Plants", the aim of this work has been to start the molecular characterization of the only sequence quoted as a high-affinity NO3- transporter in the Z. marina genome. In silico studies strongly suggest that ZosmaNTR2 encodes a plasma membrane protein that shows higher homology to NRT2.5 than to NRT2.1 transporter. This prediction is also supported by heterologous expression assays. On the other hand, the isolation of both sequences (ZosmaNRT2 and ZosmaNAR2), for the first time in a marine plant, opens the possibility to further characterization, which will provide important insights for NO3- uptake in plants under alkaline, NO3- -limited and salinized environments.

Reviewer 2 Report

The topic is interesting and authors' approach very clever. The paper can be accepted after its minor revision. Authors are engouraged to put some more effort to strengthen the novelty of their results and the potential utility of their findings for the scientific society. Some figures can be further improved

Author Response

Thank you for your comments. Manuscript has been carefully revised and minor corrections have been included through the text as follow:  

Line 2. The tittle has been changed to a more representative form for summariing the main conclusion of the work: “Molecular characterization of ZosmaNRT2, the putative sodium dependent high-affinity nitrate transporter of Zostera marina L.”

Line 7. The name of the Dpt. has been changed to Department of Botánica and Fisiología Vegetal, due to its recent designation.

Line 11. e-mail of one Vitor Amorim has been corrected: vitoramorimsilva@uma.es

Line 22: “uptake” has been replaced by “transporter”

Line 23: “as a high-affinity nitrate transporter” has been replaced by “to this function”

Lines 31 to 37, Abstract last sentences have been rewritten to strengthen the relevance of the results to improve the message as follow: “Taking together, these results suggest that Zosma70g00300.1 would encode a high-affinity nitrate transporter located at the plasma membrane, equivalent to NRT2.5 transporters. These molecular data, together with our previous electrophysiological results support that ZosmaNRT2 would have evolved to use Na+ as a driving ion, which might be an essential adaptation of seagrasses to colonize marine environments.

Line 61. Repeated “in contrast to” has been deleted

Line 66. “to mediate” has been replaced by “which meditates”

Line 67. “In addition” has been substituted by “Besides”

Line 114. “to unveil the molecular identity” has been changed by “to analyse the molecular characteristics”

Line 121. “was used” has been replaced by “was selected”

Figure 1. Colors of different motifs have been lightened, Extracellular and Intracellular compartments have been indicated as Apoplast and Cytosol, respectively, and characters for –N and –C terminal have been magnified. The role of Ser-475 on NAR2 interaction has been indicated in figure caption and discussed and referenced in the text.

Lines 136 to 138. The sequences added to the phylogenetic analysis have been cited and included in Figure 2 caption. In addition, the phylogenetic tree has been re-built using 54 sequences and light colours have been used for each clade, enlarging the numbers of the corresponding bootstrap percentage.

Selected images for Figure 3 and Figure 5 show representative pictures of ten independent assays, both figure captions have been modified. Green colour has been used for the histograms in Figure 4.

Lines 224 to 227. The requirement of co-expression of NRT2::GFP and NAR2 to visualise Hechian-strands labelled by GFP signal has been explained as follow: “Hechtian-strands labelled by GFP fluorescence were not observed in single 35S::ZosmaNRT2-GFP heterologous expression assays (Data not shown). These results support the in silico prediction for ZosmaNRT2 localization and its stabilization at the plasma membrane by NAR2, which fits to the function of ZosmaNRT2 as a NO3- transporter.”

Lines 344 to 346. Final manuscript statement has been rewritten to encourage the potential utility of the results as follow: “The isolation of both genes, for the first time in a seagrass, opens the possibility to further analysis, which would provide important insights for NO3- uptake in plants under alkaline, NO3--limited and salinized environments.”

Reviewer 3 Report

The MS appears relatively simple and it presents (indirectly) data suggesting that ZosmaNRT2 encode a putative high-affinity nitrate transporter; in fact, data concern database comparison and a stabilization by ZosmaNAR2 how it occurs in higher plants. Anyway, the MS looks nice and after a minor revision it will deserve to be published.

Minor points:

- Please correct  some spelling errors and improve a little bit the English: line 47 delete one of the “in contrast to”; line 99 ( I  am old school) delete “isolated” for “obtained” or “synthesized”; line 158 delete “cDNAs”; in the legend of Figure 5 bars is with two “r”; line 366 please delete “Isolation” for “Obtain” or “Synthesis”; line 450 replace “Garcı́a-Sánchez, M.” with “Garcı́a-Sánchez, M.J.”;

- Figure 1 legend: please add a reference for the involvement of Ser-475 in NAR2 interaction;

- Figure 1: please increase the characters for COOH and N2H; please use easily to read characters to indicate the cytosolic side and the outside / apoplast side;

- Figure 2: please use lighter, shaded colors to facilitate the reading; please introduce a space after the genera name (that should be in Italic) and the protein code/acronym;

- Figure 4: may be green histograms are more appropriate;

- Figure 5: from the legend the difference between C and D is only the magnification, so Authors should justify the difference in the brightness of the green color; moreover, in the Legend it is indicated that images were taken at 25, 26, and 28 minutes: if that is correct, it is necessary to delete “25 minutes of” at line 214.

Author Response

The MS appears relatively simple and it presents (indirectly) data suggesting that ZosmaNRT2encode a putative high-affinity nitrate transporter; in fact, data concern database comparison and a stabilization by ZosmaNAR2 how it occurs in higher plants. Anyway, the MS looks nice and after a minor revision it will deserve to be published.

 Thank you for your consideration.

Minor points:

- Please correct some spelling errors and improve a little bit the English: line 47 delete one of the “in contrast to”; line 99 (I  am old school) delete “isolated” for “obtained” or “synthesized”; line 158 delete “cDNAs”; in the legend of Figure 5 bars is with two “r”; line 366 please delete “Isolation” for “Obtain” or “Synthesis”; line 450 replace “Garcı́a-Sánchez, M.” with “Garcı́a-Sánchez, M.J.”;

 Corrected.

- Figure 1 legend: please add a reference for the involvement of Ser-475 in NAR2 interaction;

 Done, a claim for the explanation has been included in Figure 1 legend, which is given in the main text, Lines 267 to 271, Reference 36 (Ishikawa, S.; Ito, Y.; Sato, Y.; Fukaya, Y.; Takahashi, M.; Morikawa, H.; Ohtake, N.; Ohyama, T.; Sueyoshi, K. Two-component high-affinity nitrate transport system in barley: Membrane localization; protein expression in roots and a direct protein-protein interaction. Plant Biotechnol. 2009, 26, 197–205)

- Figure 1: please increase the characters for COOH and N2H; please use easily to read characters to indicate the cytosolic side and the outside / apoplast side;

Done. Colours of different motifs have been lightered, Extracellular and Intracellular compartments have been indicated as Apoplast and Cytosol, respectively, and characters for –N and –C terminal have been magnified.

- Figure 2: please use lighter, shaded colours to facilitate the reading; please introduce a space after the genera name (that should be in Italic) and the protein code/acronym;

Done. Phylogenetic tree has been re-built using 54 sequences, light colours have been used for each clade and the number size of each bootstrap percentage has been enlarged.

- Figure 4: may be green histograms are more appropriate;

Done.

- Figure 5: from the legend the difference between C and D is only the magnification, so Authors should justify the difference in the brightness of the green color; moreover, in the Legend it is indicated that images were taken at 25, 26, and 28 minutes: if that is correct, it is necessary to delete “25 minutes of” at line 214.

Done. We have repeated the experiment 10 times in total. Images are the most representatives of each these assays. Picture in A corresponded to 25 min of Mannitol treatment, picture in B to 26 min of treatment and Picture in C and D are different magnification after 28 min of treatment. Green brightness is the same in all images.

Round  2

Reviewer 1 Report

No comments